# Explicit Planning for Efficient Exploration in Reinforcement Learning

**Liangpeng Zhang**[1], **Ke Tang**[2], and **Xin Yao**[2,1*]

[1]CERCIA, School of Computer Science, University of Birmingham, U.K.
[2]Shenzhen Key Laboratory of Computational Intelligence,
University Key Laboratory of Evolving Intelligent Systems of Guangdong Province,
Department of Computer Science and Engineering,
Southern University of Science and Technology, Shenzhen 518055, China
`L.Zhang.7@pgr.bham.ac.uk, tangk3@sustc.edu.cn, xiny@sustc.edu.cn`

## Abstract

Efficient exploration is crucial to achieving good performance in reinforcement learning. Existing systematic exploration strategies (R-MAX, MBIE, UCRL, etc.), despite being promising theoretically, are essentially greedy strategies that follow some predefined heuristics. When the heuristics do not match the dynamics of Markov decision processes (MDPs) well, an excessive amount of time can be wasted in travelling through already-explored states, lowering the overall efficiency. We argue that explicit planning for exploration can help alleviate such a problem, and propose a Value Iteration for Exploration Cost (VIEC) algorithm which computes the optimal exploration scheme by solving an augmented MDP. We then present a detailed analysis of the exploration behaviour of some popular strategies, showing how these strategies can fail and spend $O(n^2md)$ or $O(n^2m + nmd)$ steps to collect sufficient data in some tower-shaped MDPs, while the optimal exploration scheme, which can be obtained by VIEC, only needs $O(nmd)$, where $n, m$ are the numbers of states and actions and $d$ is the data demand. The analysis not only points out the weakness of existing heuristic-based strategies, but also suggests a remarkable potential in explicit planning for exploration.

## 1 Introduction

In reinforcement learning (RL), exploration plays a key role in deciding the quality of data and thus has a direct impact to the overall performance. Simple exploration strategies such as $\varepsilon$-greedy may need exponentially many steps to find a (near-)optimal policy [1]. On the other hand, more systematic exploration strategies (R-MAX, UCRL, MBIE and their variants) have far promising theoretical performance guarantees (see e.g. [2, 3, 4, 5]). Recently, some of these systematic strategies have been successfully generalised and applied to deep reinforcement learning, achieving good performance in domains that are known to be hard to explore, such as Montezuma's Revenge [6, 7].

Systematic exploration strategies are carefully designed to ensure that sufficient data is collected for every unknown states, so that the chance of converging to undesirable policies due to ignorance is controlled. Unfortunately, the actual data collection process is less carefully executed, in the sense that these strategies choose actions simply by maximising some predefined heuristics. When the design of such heuristics does not match the properties of the learning problem well, an excessive amount of less useful data will be collected due to revisiting well-explored states/actions.

A straightforward example is as follows. Suppose both a nearby $\text{Area}_1$ and a distant $\text{Area}_2$ need to be explored. The transition dynamics makes it easy to travel from $\text{Area}_2$ to $\text{Area}_1$, but trying to move from $\text{Area}_1$ to $\text{Area}_2$ sends the agent back to $\text{Area}_1$ with high probability. Clearly, exploring in the order of $\text{Area}_2 \rightarrow \text{Area}_1$ is better than $\text{Area}_1 \rightarrow \text{Area}_2$, since the latter wastes additional time in trying to travel to $\text{Area}_2$ from $\text{Area}_1$ which leads to excessive data being collected in $\text{Area}_1$. However, most systematic strategies choose to explore $\text{Area}_1$ first because it is nearer than $\text{Area}_2$ and thus has a higher heuristic score. We call this a *distance trap*.

Our analysis in this paper points out that there exist cases where these heuristic-based strategies need either $O(n^2md)$ or $O(n^2m + nmd)$ steps to collect sufficient data, while an optimal exploration scheme only needs $O(nmd)$, where $n$, $m$, and $d$ denote number of states, number of actions, and the minimum amount of data to be obtained at each state-action pair, respectively. Since $n$ is usually very large in real-world problems, this result indicates that a significant amount of steps can be wasted by the heuristic-based strategies due to their careless execution of data collection. It also suggests that *explicit planning for exploration* can be highly beneficial for improving learning efficiency.

The contributions of this paper are as follows.

1. Formulate the planning for exploration problem as an augmented undiscounted Markov decision process and show that the optimal exploration scheme can be discovered by solving the Bellman optimality equations for exploration costs.

2. Propose a Value Iteration for Exploration Cost (VIEC) algorithm for finding the optimal exploration scheme.

3. Point out two weaknesses of existing systematic exploration strategies: (a) distance traps and (b) reward traps, and use tower MDPs as examples to give a concrete explanation about how existing strategies can fail and need $O(n^2md)$ or $O(n^2m + nmd)$ steps while the optimal exploration scheme needs only $O(nmd)$ steps to fulfil the same exploration demand.

## 2 Preliminaries

In this paper we follow the common formulation of reinforcement learning [8] in which $M = (\mathcal{S}, \mathcal{A}, P, R, \gamma)$ represents a finite discounted Markov decision process (MDP) with set of states $\mathcal{S}$, set of actions $\mathcal{A}$, transition probability function $P$, reward function $R$, and discount factor $\gamma$. Unless otherwise stated, we use $n$ and $m$ to denote the number of states and actions of an MDP. A policy is denoted $\pi$ and its value functions are denoted $V^\pi(s)$ and $Q^\pi(s, a)$, while for optimal policy we write $\pi^*$, $V^*$ and $Q^*$, which by definition satisfy $V^*(s) = \max_\pi V^\pi(s)$ and $Q^*(s, a) = \max_\pi Q^\pi(s, a)$ for all $s \in \mathcal{S}$ and $a \in \mathcal{A}$. If exact information about $M$ is available, then $\pi^*$ can be obtained by solving the Bellman equations $V^*(s) = \max_a(\mathbb{E}[R(s, a)] + \gamma \sum_{s'} P(s'|s, a)V^*(s'))$ or $Q^*(s, a) = \mathbb{E}[R(s, a)] + \gamma \sum_{s'} P(s'|s, a) \max_{a'} Q^*(s', a')$ using Value Iteration algorithm [9].

In reality $M$ is often unknown and needs to be estimated from the data collected during learning. A straightforward way is to use $\hat{P}(s'|s, a) = N(s, a, s')/N(s, a)$ and $\hat{R}(s, a) = C(s, a)/N(s, a)$ as estimates of $P(s'|s, a)$ and $\mathbb{E}[R(s, a)]$, where $N(s, a)$ and $N(s, a, s')$ indicate the occurrences of choice $(s, a)$ and transition $(s, a, s')$ and $C(s, a)$ is the sum of the rewards collected at $(s, a)$. As $N(s, a) \rightarrow \infty$ at all $(s, a)$, this model $\hat{M}$ of $M$ converges in probability to the true $M$, and thus we can eventually obtain $\pi^*$ of $M$ from $\hat{M}$. Such process is called model-based RL.

Researches on systematic exploration are often based on model-based RL, so that the quality of learning is mostly decided by their exploration strategies. This paper follows this idea and limits its scope to the model-based case, but its general suggestion (explicit planning for exploration can be beneficial) is also applicable to model-free RL.

## 3 Formulation of the Planning for Exploration Problem

### 3.1 Data demands

Since the goal of learning is to find out a sufficiently good policy rather than to have an extremely accurate estimate of $V$ or $Q$, a finite amount of data is often sufficient for the purpose. Various researches have shown that by applying Hoeffding's or Chernoff's inequalities, the minimum amount

of data needed at each state-action pair for guaranteeing certain learning quality can be derived. For example, [2, 10] proved that some $O(\frac{1}{\varepsilon^2(1-\gamma)^4}(n+\ln\frac{nm}{\delta}))$ data for each state-action pair is sufficient for R-MAX to be $(\varepsilon, \delta)$-PAC, while [4] proved that for MBIE it is $O(\frac{1}{\varepsilon^2(1-\gamma)^4}(n + \ln\frac{nm}{\varepsilon(1-\gamma)\delta}))$, where $n$ and $m$ are the number of states and actions.

In practice, the theoretical demands of this kind are still likely to be excessive (see e.g. [11, 12, 13]), and users usually have to specify how much data to be collected based on their domain knowledge or trial-and-error. Whichever the case, the main idea is that such data demands are given (either directly or indirectly) by the parameter settings prior to the actual learning process, and thus can be used to make plans for more efficient exploration. The formal definition of data demands is as follows.

**Definition 3.1** *In an MDP with $n$ states and $m$ actions, a demand matrix $D$ is an $n \times m$ matrix in which entry $D[s, a] = k \geq 0$ indicates that at least $k$ more data should be collected for state-action pair $(s, a)$ during learning.*

We write $D_t$ to indicate the demand matrix at time $t$ during learning. After some action $A_t$ is executed at some state $S_t$, the corresponding entry in the demand matrix should be subtracted by 1 unless it is already 0, while other entries remain unchanged, that is,

$$D_{t+1}[s, a] = \begin{cases} \max\{0, D_t[s, a] - 1\} & (s, a) = (S_t, A_t) \\ D_t[s, a] & \text{otherwise.} \end{cases}$$

For convenience, we define the demand reduction function $H$ as follows:

$$H(D; s, a) := \begin{cases} D - \boldsymbol{e}_{s,a} & D[s, a] > 0 \\ D & D[s, a] = 0, \end{cases}$$

where $\boldsymbol{e}_{s,a}$ is an $n \times m$ matrix filled with 0 except for the only nonzero entry $\boldsymbol{e}_{s,a}[s, a] = 1$. Then we can express the change of $D_t$ after $(S_t, A_t)$ simply as $D_{t+1} = H(D_t; S_t, A_t)$.

The demand space (the set of all possible demand matrix) of an MDP is denoted $\mathcal{D}$. It is reasonable to assume that the demands at every state-action never exceed some sufficiently large positive integer $d$, thus the size of demand space is at most $(d + 1)^{nm}$.

**Remark.** Readers may wonder how to find out the "optimal" demand matrix (that has e.g. the least total demand) for a given learning task. Such matrix can only be obtained with full knowledge of the MDP, and thus is impractical to obtain in reality. Our point is that given *any* demand matrix, the exploration efficiency can be improved via planning. It is achieved by minimising the amount of data collected beyond the specified demand (i.e. optimal exploration scheme, see next section) rather than choosing a better demand matrix, and thus the optimality of demand matrices is not the main concern of this paper.

## 3.2 Planning for exploration

Demand matrix $D$ indicates how much data is sufficient for obtaining a good policy, and we are interested in collecting all this required amount of data with the number of steps as small as possible, since this means that the least amount of unnecessary data is collected beyond $D$. The exploration behaviour of a learning agent can be described as an exploration scheme, while its exploration cost is the expected number of steps needed to fulfil all the demands, defined formally as follows.

**Definition 3.2** *An exploration scheme $\psi$ is a mapping $\mathcal{D} \times \mathcal{S} \mapsto \mathcal{A}$, where $\psi(D; s) = a$ indicates that action $a$ should be taken at state $s$ when the demand matrix is $D$.*

**Definition 3.3** *The exploration cost $C^\psi(D; s, a)$ is the expected time $t$ that the current demand $D_t$ first becomes the all-zero matrix $\mathbf{0}$ by starting from $(s, a)$ and following $\psi$, i.e. $C^\psi(D; s, a) := \mathbb{E}[\inf\{t : D_{t+1} = \mathbf{0} | D_1 = D, S_1 = s, A_1 = a, A_k = \psi(D_k; S_k) \ \forall k > 1\}]$.*

Given MDP $M$ and exploration scheme $\psi$, the interaction process becomes a Markov process with augmented state space $\mathcal{D} \times \mathcal{S}$ and transition probability $\Pr(D', s'|D, s) = P(s'|s, \psi(D; s))$ for $D' = H(D; s, \psi(D; s))$ and 0 otherwise. As for the exploration cost, by definition when $D = \mathbf{0}$ we have $C^\psi(D; s, a) = 0$ for any $(s, a)$. Any step after $D_t = \mathbf{0}$ will not result in any exploration

cost, while each step before reaching $D_t = \mathbf{0}$ will increase the cost by 1 uniformly. Therefore, the planning for exploration problem is an augmented undiscounted MDP, and the following Bellman equation holds for the exploration cost:

$$C^\psi(D; s, a) = \begin{cases} 1 + \sum_{s' \in \mathcal{S}} P(s'|s, a)\, C^\psi\big(H(D; s, a); s', \psi\big(H(D; s, a); s'\big)\big) & D \neq \mathbf{0} \\ 0 & D = \mathbf{0}. \end{cases} \quad (1)$$

Let $\Psi$ be the set of all possible exploration schemes for a given MDP. Since less exploration cost is more desirable, the definition for the optimal scheme is as follows.

**Definition 3.4** *An optimal exploration scheme $\psi^* \in \Psi$ is the one that satisfies $C^{\psi^*}(D; s, a) = \min_{\psi \in \Psi} C^\psi(D; s, a)$ for any $D \in \mathcal{D}$, $s \in \mathcal{S}$ and $a \in \mathcal{A}$.*

For convenience we write the optimal exploration cost $C^{\psi^*}$ simply as $C^*$. In strongly connected MDPs, it can be shown that similar to optimal value functions $Q^*$ and $V^*$, the optimal exploration cost $C^*$ exists and is unique. In MDPs that are not strongly connected, on the other hand, there exist cases where some demands are not satisfiable. For example, in an MDP with two states $\{s_1, s_2\}$ and one action $a$ which transits the agent to $s_2$ with probability 1 from both $s_1$ and $s_2$, a demand $D[s_1, a] > 1$ can never be satisfied and will lead to an infinite exploration cost. However, as discussed in Section 3.1, since users more or less have control to the exploration demands, in the rest of this paper we assume that they do not assign unsatisfiable demands and thus $C^*$ exists.

### 3.3   Computing $\psi^*$

By combining Equation 1 with Definition 3.4 we get the Bellman optimality equation for $C^*$:

$$C^*(D; s, a) = \begin{cases} 1 + \sum_{s' \in \mathcal{S}} P(s'|s, a)\, \min_{a' \in \mathcal{A}} C^*(H(D; s, a); s', a') & D \neq \mathbf{0} \\ 0 & D = \mathbf{0}. \end{cases} \quad (2)$$

Since this equation has structure similar to the original Bellman optimality equation for $Q^*$ and $V^*$, we can modify Value Iteration to compute $C^*$. Note that $H(D; s, a) \leq D$ for any $D$, $s$ and $a$, given an input demand matrix $D_{\text{in}}$, we can easily arrange all $k = \prod_{s,a}(D_{\text{in}}[s, a] + 1)$ demand matrices satisfying $D \leq D_{\text{in}}$ by topological ordering, i.e. $D_{(0)} = (0...0; ...; 0...0)$, $D_{(1)} = (0...0; ...; 0...1)$, ..., $D_{(k-1)} = D_{\text{in}}$, and compute $C^*$ from $D_{(0)}$ to $D_{(k-1)}$ to avoid extra iterations on $D$.

The pseudocode of the Value Iteration for Exploration Cost (VIEC) is presented in Algorithm 1, where $U(D; s) := \min_a C(D; s, a)$, which plays a role similar to $V(s)$ in computing $Q(s, a)$.

---

**Algorithm 1** Value Iteration for Exploration Cost (VIEC)

---

**Input**: Demand matrix $D_{\text{in}}$, transition $P$
**Output**: Exploration scheme $\psi$
1:  Initialise all $C(D; s, a) = 0$, $U(D; s) = 0$
2:  **for** $i = 1$ to $\prod_{s,a}(D_{\text{in}}[s, a] + 1) - 1$ **do**
3:     **repeat**
4:        $\Delta = 0$
5:        **for** $s \in \mathcal{S}$ **do**
6:           **for** $a \in \mathcal{A}$ **do**
7:              $c = 1 + \sum_{s'} P(s'|s, a)\, U\big(H(D_{(i)}; s, a); s'\big)$
8:              $\Delta = \max\{\Delta, |C(D_{(i)}; s, a) - c|\}$
9:              $C(D_{(i)}; s, a) = c$
10:          $U(D_{(i)}; s) = \min_a C(D_{(i)}; s, a)$
11:    **until** $\Delta <$ threshold
12: Output $\psi$ such that $\psi(D; s) = \text{argmin}_a C(D; s, a)$

---

Similar to the original Value Iteration, with a sufficiently small stopping threshold in Line 11, $C$ converges to $C^*$ and thus the output $\psi \to \psi^*$. The proof can be obtained straightforwardly from the convergence proof of original Value Iteration and will not be elaborated here.

VIEC needs to iterate over $\prod_{s,a}(D_{\text{in}}[s,a] + 1) = O(d^{nm})$ demand matrices and is not computationally efficient in practice. Unfortunately, this is unavoidable for computing $\psi^*$ because even in the simplest case with deterministic transitions and demands no more than 1, it is a Rural Postman Problem [14] which is NP-hard, thus solving in polynomial time is impossible unless P=NP. Nevertheless, an approximation to $\psi^*$ might be sufficient for the purpose, and we leave this to the future work.

In most RL settings, the transition function $P$ is not known to the learning algorithm in prior. In this case, one possible choice is to use estimated transition $\hat{P}$ instead, following an iterative process shown in Algorithm 2. In this case, output $\psi$ of VIEC is an optimal exploration scheme for the environment model $\hat{M}$ rather than the true environment $M$. With more data been collected, $\hat{M}$ gets closer to $M$ and thus $\psi$ becomes closer to the true $\psi^*$. While a $\psi$ improving over time is surely not as good as $\psi^*$, it is still better than never conduct planning, and thus Algorithm 2 should provide a relatively efficient way of exploration in general.

---

**Algorithm 2** Model-based RL with Planning for Exploration

**Input**: Initial demand $D_1$
**Output**: Policy $\pi$

1: Initialise $\hat{P}, \hat{R}$ randomly or based on prior knowledge
2: $\psi = \text{VIEC}(D_1, \hat{P})$
3: **repeat**
4:     Collect data by following $\psi$
5:     Update $\hat{P}, \hat{R}, D_t$ using collected data
6:     Update $\psi$ using $\text{VIEC}(D_t, \hat{P})$
7:     Update $\pi$ using Value Iteration$(\hat{P}, \hat{R})$
8: **until** $D_t = \mathbf{0}$ or $\pi$ is sufficiently good

---

## 4 When and How Heuristics Fail and Explicit Planning Helps

Systematic exploration strategies choose action by maximising some predefined heuristics $\tilde{Q}(s,a)$, which is prone to the traps as follows. Suppose at current state $S_t$ and demand $D_t$ there are actions $a_1, a_2$ satisfying $C(D_t; S_t, a_1) < C(D_t; S_t, a_2)$, so one should choose $a_1$ over $a_2$.

**Distance traps.** Let the nearest (in terms of expected number of steps to arrive) to-be-explored state after taking $a_1$ and $a_2$ be $s'$ and $s''$ respectively. If $s''$ is closer to $S_t$ than $s'$, then the uncertainty of $s''$ is less discounted than $s'$ in $\tilde{Q}$, resulting in $\tilde{Q}(S_t, a_1) < \tilde{Q}(S_t, a_2)$ and thus $a_2$ is picked.

**Reward traps.** Let the reward (or expected return) of taking $a_1$ and $a_2$ be $r'$ and $r''$, respectively. Then $r' < r''$ can lead to $\tilde{Q}(S_t, a_1) < \tilde{Q}(S_t, a_2)$ and thus $a_2$ is picked.

These traps can appear in any MDP and significantly reduce the efficiency of heuristic-based strategies. To present this more clearly and intuitively, we introduce a class of MDPs called tower MDPs, analyse the behaviours and exploration costs of several typical exploration strategies and the optimal exploration scheme in tower MDPs, and then discuss the implication of the results.

### 4.1 Tower MDPs

A tower MDP of height $h$ has two groups of states, namely upward states $s_1, ..., s_h$ and downward states $s'_1, ..., s'_h$. The total number of states is $n = 2h$. The agent always starts interaction from $s_1$. An example with height $h = 5$ is shown in Figure 1.

The transitions are deterministic in tower MDPs. Each upward state $s_k$ has an action $a'$ that transits the agent to $s'_k$ (dashed arrows in Figure 1), and also an action $a$ that transits to $s_{k+1}$ if $k < h$ (solid arrows). Each downward state $s'_k$ is an $m$-armed bandit, which has $m$ actions $a_1, ..., a_m$ that yield rewards following some predefined distributions and transit the agent to $s'_{k-1}$ $(k > 1)$ or $s_1$ $(k = 1)$ (collectively drawn as the double arrows in Figure 1).

To find out the optimal policy, the agent has to collect data in these bandits for information about their reward distributions. For simplicity we assume that the initial demands at each of these $m$-armed

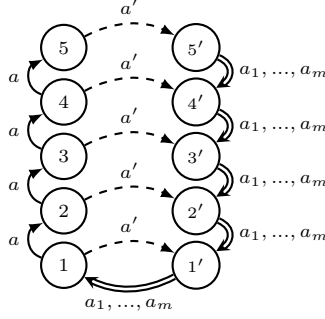

Figure 1: A tower MDP of height $h = 5$. Each double arrow represents an $m$-armed bandit.

bandits are uniformly set to some $d > 0$. As for $a$ and $a'$ in upward states, since there is no uncertainty at all, their initial demands are set to 0.

## 4.2 Optimal exploration scheme

In a tower MDP of height $h$ with $m$-armed bandits in downward states, it is easy to see that the optimal scheme to collect $d$ data at each arm is to repeatedly take the closed path $[s_1 s_2 ... s_h s'_h ... s'_1 s_1]$. Each time taking this path, the demand of one arm at every downward state is reduced by 1, and thus it needs to be repeated $md$ times to collect all the data required. Since the length of this path is $2h$, the optimal exploration scheme needs $2hmd = O(nmd)$ steps to fully satisfy the initial demands.

## 4.3 $\varepsilon$-greedy

Although $\varepsilon$-greedy is already well-known for its lack of efficiency, it is nevertheless interesting to see how it performs in tower MDPs. Let the bandit in $s'_1$ gives a reward of 1 with probability 1 on all of its $m$ arms, $a_m$ in $s'_h$ gives reward $10^{10}$ with probability 0.01 and reward 0 otherwise, while all other bandits/arms give zero reward. At the beginning of learning, $\varepsilon$-greedy does not know any of these rewards, and thus has a 50-50 chance to choose between going to state $s_2$ and $s'_1$. If it chooses $s_2$, then it has another 50-50 chance between $s_3$ and $s'_2$, and so on. Therefore, the probability it arrives at $s_h$ without visiting any of $s'_1, ..., s'_{h-1}$ is $0.5^{h-1}$. If it ever goes to any of state $s'_1, ..., s'_{h-1}$ before arriving at $s_h$, which happens with probability $1 - 0.5^{h-1}$, it will be aware of the reward at $s'_1$, and thereafter be trapped to going to $s'_1$ as often as possible. Whenever it gets back to $s_1$, it only has probability $(0.5\varepsilon)^{h-1}$ to randomly wanders into $s'_h$.

Therefore, the average number of steps $\varepsilon$-greedy spends to visit $s'_h$ once is $0.5^{h-1} \cdot 2h + (1 - 0.5^{h-1})O(\frac{h}{(0.5\varepsilon)^{h-1}}) = O(n2^n)$ if $\varepsilon$ is seen as a constant. Since it needs to visit $s'_h$ $(md)$ times to fully fulfil the demands, the exploration cost of $\varepsilon$-greedy is $O(nmd\,2^n)$.

## 4.4 R-MAX

R-MAX [15] is one of the first systematic strategies that are proved to have polynomial sample complexity upper bounds [2, 10]. Many exploration strategies are designed based on R-MAX and have similar performance guarantees, including Delayed Q-learning [16], MoR-MAX [17], V-MAX [18], and ICR [13], just to name a few.

R-MAX works as follows. When a state-action pair has a positive demand to fulfil, it is labelled "unknown" and its estimated value $\tilde{Q}(s, a)$ is set to $V_{\max} := \frac{R_{\max}}{1-\gamma}$, where $R_{\max}$ is the maximum possible reward. If its demand is already 0, then it is labelled "known" and the algorithm uses the Bellman equation to estimate its $\tilde{Q}(s, a)$. R-MAX always chooses the action with maximum $\tilde{Q}(s, a)$.

In tower MDPs, all actions in downward states are initially "unknown" and thus their $\tilde{Q} = V_{\max}$ at the beginning of learning. Let the bandits at all states except $s'_h$ give zero reward, while the bandit at $s'_h$ gives reward $R_{\max} = 1$ with probability 0.1 and reward 0 otherwise for all arms. Under such setting, R-MAX will not be aware of any positive rewards until $s'_h$ is explored. It can be shown recursively

that at this stage of learning, at any upward state $s_k$, R-MAX will choose $a'$ to go to $s'_k$ rather than $a$ that goes to $s_{k+1}$. Concretely, at $s_h$, the only choice is $a'$ which leads to "unknown" actions in $s'_h$, thus has $\tilde{Q}(s_h, a') = \gamma V_{\max}$. At state $s_{h-1}$, going to $s_h$ has value $\tilde{Q}(s_{h-1}, a) = \gamma \tilde{Q}(s_h, a') = \gamma^2 V_{\max}$, while going to $s'_{h-1}$ has $\tilde{Q}(s_{h-1}, a') = \gamma V_{\max} > \tilde{Q}(s_{h-1}, a)$, thus R-MAX will choose $a'$ at $s_{h-1}$ as well. The same happens at every state from $s_{h-1}$ down to $s_1$. Since the agent starts from state $s_1$, R-MAX will stick to $[s_1 s'_1 s_1]$ until all $a_1, ..., a_m$ at $s'_1$ are tried $d$ times and become "known".

After collecting sufficient data at state $s'_1$, $\tilde{Q}(s_1, a)$ drops greatly from $\gamma V_{\max}$ to $\gamma^4 V_{\max}$ and R-MAX starts choosing $a$ at $s_1$. Since $\tilde{Q}$ at states other than $s_1$ and $s'_1$ remain unchanged, $s'_2$ is the next target of exploration due to having the least discount in $\tilde{Q}$. This leads to a behaviour of taking $[s_1 s_2 s'_2 s'_1 s_1]$ to collect at $s'_2$, then $[s_1...s_3 s'_3...s'_1 s_1]$ for $s'_3$, and so on, and finally $s'_h$. The exploration cost of such process is $2md + 4md + ... + (2h)md = h(h+1)md = O(n^2 md)$.

## 4.5 Interval estimation

Interval estimation (IE) based exploration strategies utilise statistical methods to create confidence intervals (CIs) for the estimated models or state/action values. CIs computed by this type of strategies usually take the form of $X(s, a) \pm \frac{\beta}{\sqrt{N(s,a)}}$, where $X(s, a)$ is the variable being estimated, $\beta$ is a parameter, and $N(s, a)$ is the amount of data collected at $(s, a)$. Clearly, state-action pairs with less data have longer CIs, and vice versa. Estimated variable $X(s, a)$ can be transition probability, reward, or state/action values. When choosing actions, the action with highest estimated value among all possible MDP models that lie within the CIs is selected.

In this section we take MBIE-EB as example to show how IE-based strategies can be tricked to make inferior decisions. In MBIE-EB, action values are estimated using $\tilde{Q}(s, a) = \tilde{R}(s, a) + \gamma \sum_{s'} \hat{P}(s'|s, a) \max_{a'} \tilde{Q}(s', a')$, where $\tilde{R}(s, a) := \hat{R}(s, a) + \frac{\beta}{\sqrt{N(s,a)}}$. Since $N(s, a) = 0$ leads to division by zero, in the following analysis we assume that they all start with 1. At each step the action with highest $\tilde{Q}(S_t, a)$ is executed, thus $\tilde{Q}$ is the heuristic used in MBIE-EB.

We start our analysis with the simplest case $m = 1$ where all bandits in the tower MDP is one-armed. The expression of $\tilde{Q}$ can be obtained by solving the Bellman equation. Note that although max operator is involved on all state-action pairs, the algorithm is essentially choosing between paths $[s_1...s_j s'_j...s'_1 s_1]$ with different $j$. Let $\tilde{Q}_j$ be $\tilde{Q}$ for the $j$-th path, $\tilde{R}_j$ be $\tilde{R}$ for the bandit at $s'_j$, and $N_j$ be $N(s, a)$ at that bandit, then we have $\tilde{Q}_j = \frac{\gamma^j}{1-\gamma^{2j}} \sum_{i=1}^{j} \gamma^{j-i} \tilde{R}_i$.

Let the actual reward of the bandits be the same as the settings used in the analysis of R-MAX. At the beginning of learning $\tilde{R}_j = \beta/\sqrt{N_j} = \beta$, thus $\tilde{Q}_j = \frac{\beta}{1-\gamma}(1 - \frac{1}{1+\gamma^j})$. Clearly $\tilde{Q}_1 > \tilde{Q}_2 > ... > \tilde{Q}_h$, thus MBIE-EB starts with path $[s_1 s'_1 s_1]$, which increases $N_1$ and reduces $\tilde{R}_1$.

The expression $\tilde{Q}_j = \frac{\gamma^j}{1-\gamma^{2j}} \sum_{i=1}^{j} \gamma^{j-i} \tilde{R}_i$ shows that $\tilde{Q}_j$ with larger $j$ has a greater discount $\gamma^{j-i}$ on $\tilde{R}_i$, and thus exploring $s'_1$ reduces $\tilde{Q}_j$ less for larger $j$. Therefore, $\tilde{Q}_2$ will eventually surpass $\tilde{Q}_1$ and MBIE-EB moves to exploring $s'_2$, then $s'_3$, and so on, leading to an exploration behaviour similar to R-MAX, but lingers less at the same state than R-MAX. A smaller discount factor $\gamma$ leads to a larger gap between different $\tilde{Q}_j$, which then leads to a slower pace for MBIE-EB to move upward. In the case where MBIE-EB only lingers exactly once at each level of the tower, it will take path $[(s_1 s'_1)(s_1 s_2 s'_2 s'_1)(s_1 s_2 s_3 s'_3 s'_2 s'_1)...]$ until $s_h$ is reached. Thereafter $\tilde{Q}_h$ will always be the largest, and thus the remaining demand will be fulfilled through repeating $[s_1...s_h s'_h...s'_1 s_1]$. Such behaviour has exploration cost $(2 + 4 + ... + 2h) + 2h(d - 1) = h(h + 1) + 2h(d - 1) = O(n^2 + nd)$ steps. For the sake of space we skip the full derivation here[2], but a $\gamma < \frac{\sqrt{5}-1}{2} \approx 0.618$ is sufficient to make sure that MBIE-EB will perform as bad as this[3].

| Strategy | Exploration cost | Weakness |
|---|---|---|
| Optimal scheme | $O(nmd)$ | - |
| $\varepsilon$-greedy | $O(nmd\,2^n)$ | Distance, reward |
| R-MAX | $O(n^2md)$ | Distance |
| Interval estimation | $O(n^2m+nmd)$ | Distance, reward |

Table 1: Summary of results on tower MDPs.

In the case of $m \geq 2$ where there are 2 or more arms in each bandit, $\tilde{R}_j$ in the expression of $\tilde{Q}_j$ becomes the maximum of $\frac{\beta}{\sqrt{N_{j,k}}}$ where $N_{j,k}$ is the number of data at the $k$-th arm at state $s'_j$. As a result, the same pattern as in $m = 1$ is repeated $m$ times for the case of $m \geq 2$, and thus the total exploration cost is $O(n^2m + nmd)$.

Note that MBIE-EB and other IE-based exploration strategies also take $\hat{R}(s,a)$ into consideration when choosing actions, and thus can be further tricked by a deceiving setting of true reward $R(s,a)$. For example, if the setting of rewards in Section 4.3 is used, then more weight will be put into $\tilde{R}_1$, which gives $\tilde{Q}_j$ with smaller $j$ more advantage due to having smaller discount on $\tilde{R}_1$. As a result, MBIE-EB will stay at lower levels more often and thus will have a worse exploration cost than above.

## 4.6 Discussion

Table 1 sums up the results of the analysis. As can be seen from the exploration cost column, $\varepsilon$-greedy is clearly inferior to the rest for being exponential to the number of states $n$. MBIE-EB is seemingly better than R-MAX, but since in reality it often happens that $n \gg d$, the difference between the two can be small, and both are far worse than the optimal scheme which is only $O(nmd)$. Such results suggest that explicit planning for exploration can be highly beneficial when the state space is large.

It is interesting to compare the exploration costs with sample complexity bounds, a well-studied exploration efficiency metric. R-MAX and MBIE-EB have sample complexity upper bounds $O(n^2m)$ (ignoring other factors and logarithms) [2, 4], which is similar to the exploration costs $O(n^2md)$ on $n$ and $m$. However, a variant of R-MAX called MoR-MAX is known to have sample complexity $O(nm)$ [17], yet its exploration cost in tower MDPs is still $O(n^2md)$ due to having exactly the same behaviour as R-MAX. This might explain why sample complexity is usually not a good indicator of practical exploration efficiency.

The "distance" and "reward" in the weakness column of Table 1 refers to the distance traps and reward traps mentioned at the beginning of Section 4. A longer distance makes $\varepsilon$-greedy visit states in the higher levels via random walk less often, while for R-MAX and IE algorithms a longer distance leads to more discount and thus a lower heuristic score $\tilde{Q}$ for the states in the higher levels. Reward traps lure both $\varepsilon$-greedy and IE to the lower-level states, while R-MAX is more resistant to it due to using $V_{\max}$ in computing $\tilde{Q}$. The optimal scheme is the result of minimising undiscounted exploration cost and is affected by neither traps.

Tower MDPs in the above analysis only use deterministic transitions for simplicity. In non-deterministic cases, the negative impact of distance traps can be even more severe due to transition probabilities amplifying the gaps in average distances. For example, if transiting from state $s_1$ to $s'_1$ by taking $a'$ has probability 1, while taking $a$ at $s_1$ has probability 0.5 to go to $s_2$ and probability 0.5 to stay in $s_1$, then the gap between $\tilde{Q}_1$ and $\tilde{Q}_2$ becomes larger and IE algorithms will take path $[s_1 s'_1 s_1]$ more often, increasing the exploration cost.

MDPs in reality may not have the same structure as tower MDPs, but the distance traps and the reward traps discussed above can happen in any types of MDPs. It is possible that in some easier cases the difference between the optimal scheme and heuristic-based strategies is not as large as $O(nmd)$ vs. $O(n^2md)$, but in domains where millions of data is required for obtaining an acceptable policy, even a difference in constant factor can be practically significant.

**Remark on the reward trap**. One may argue that whether or not the reward function actually acts as a trap is problem-dependent, and there are cases where being trapped by the rewards is actually desirable due to it leading to an early convergence to good policies.

This is partly true, but one should also consider the fact that it is very difficult, if not impossible, to design a reward function that simultaneously leads to both good policies and good exploration behaviours. If the way being trapped does not coincide with the (near-)optimal policies, algorithms like MBIE eventually deviate from the current behaviour and restart exploration. In the early stages of learning, since the data is still lacking, such situation can occur frequently, resulting in the whole learning process being prolonged and the total reward reduced.

Therefore, even in the case where the total reward during learning is of concern, ignoring it in the early stages of the learning process and seeking for a more efficient exploration behaviour can be beneficial in the long run.

## 5    Conclusion and Future Work

In this paper we have formulated the planning for exploration problem as solving augmented MDPs, and provided the Bellman optimality equation for exploration costs. We have proposed a Value Iteration for Exploration Cost (VIEC) algorithm which computes the optimal exploration scheme given full knowledge of MDP, and a model-based RL method with planning-for-exploration component integrated. We have presented a detailed study of exploration behaviours of several popular exploration strategies. The analysis exposes the weakness of these heuristic-based strategies and suggests a remarkable potential in planning for exploration.

A possible direction for future work is to find a fast and sufficiently good approximate to VIEC. As we pointed out in Section 3.3, since the demand space is exponential in the number of states, applying VIEC directly can be computationally expensive in practice. Techniques such as Prioritized Sweeping [19] may help reduce the computation involved, thus make VIEC more practically useful.

Another direction is to design better heuristic-based exploration strategies that can handle the distance and reward traps discussed in Section 4 better. Although by No Free Lunch theorem [20] no heuristic can perform universally better than others, it is nevertheless useful to have a larger toolbox of easy-to-compute heuristics that can cope with different types of MDPs.

### Acknowledgments

This work was supported by EPSRC (Grant Nos. EP/J017515/1 and EP/P005578/1), the Royal Society (through a Newton Advanced Fellowship to Ke Tang and hosted by Xin Yao), the Program for Guangdong Introducing Innovative and Enterpreneurial Teams (Grant No. 2017ZT07X386), Shenzhen Peacock Plan (Grant No. KQTD2016112514355531) and the Program for University Key Laboratory of Guangdong Province (Grant No. 2017KSYS008).

## Footnotes

*The corresponding author.

[2] The proof will be given in the online supplementary material.

[3] Note that $\gamma < 0.618$ can effectively be achieved by inserting additional dummy states into all transitions, e.g. if $\gamma = 0.9$, by inserting 4 states between all transitions the discount becomes $0.9^5 = 0.59 < 0.618$.

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
