[Reviews · NeurIPS 2019]

Reviewer 1



This paper introduces the interesting idea of demand matrices to more efficiently do pure exploration. Demand matrices simply specific the minimum number of times needed to visit every state-action pair. This is then treated as an additional part of the state in an augmented MDP, which can then be solved to derive the optimal exploration strategy to achieve the specified initial demand. While the idea is interesting and solid, there are downsides to the idea itself and some of the analysis in this paper that could be improved upon. There are no theoretical guarantees that using this algorithm with a learned model at the same time will work. It seems possible that if we specify a large enough lower bound on the entries in the demand matrix, then a PAC bound might be able to be proven that will guarantee that eventually everything will be explored properly. It would greatly help the paper if such a theorem was proven. While the concept behind distance traps is sound, the idea that reward traps are bad is not as clear. R-MAX/MBIE-like algorithms explicitly consider the reward function, because they are designed to eventually converge to the optimal policy and automatically trade-off between exploration and exploitation. This algorithm is doing pure exploration, and thus perhaps a better comparison would be the exploration component of the E^3 which had explicit exploration and exploitation components. You could design domains where MBIE can outperform VIEC, for example the Six Arms domain where as soon as MBIE finds the best arm it says there and no longer tries the other arms; VIEC would continue to try other arms and thus be less efficient. It would be good to clearly make this distinction of purpose in the paper. The final large drawback is the computational cost. The augmented MDP's state space is an exponential blowup of the original state space, making the new problem much more difficult to solve. It would be useful, though perhaps very difficult, to try to derive some simpler approximations of this idea. Overall the idea is interesting, but there is a lack of more in-depth analysis. There is no theory on whether learning without a known model works, nor any considering for how to make this practical. There should be more analysis in at least one of these directions. ** Update after author response ** After thinking more, I have increased my score to a 6, because of the potential of future research with the demand matrix. However I would like to better clarify my view of this paper and hopefully the author can better address these points in the final copy. PAC theorem: I fully do not expect the authors to be able to add this during the rebuttal process, and it was a mistake on my part to put this in the improvements as it was impractical. However I hope that in the future this can be one focus. In the author response, they mention that PAC focuses on worst case, and this is indeed exactly why I mentioned it. Right now, without given a model of the environment, this is no guarantee that in the worst case, some variation of VIEC that tries to learn the model simultaneously will not just fail to satisfy the demand matrix i.e. Algorithm 2 has no guarantee that it will not just sometimes fail. While PAC is not a good analysis tool in comparing exploration, it at the very least is a safety net that says that your algorithm will not outright fail. Reward traps: I brought up the SixArms example as where MBIE is faster than VIEC, and the authors have countered with the fact that given an appropriate demand matrix, VIEC is more optimal. Yes, this is true, but whole point of MBIE is that you do not need to know the optimal demand matrix. Without prior knowledge, you do not know that one of the arms is optimal, and hence you have no way of knowing the optimal demand matrix. The alternative is to use a generic demand matrix such as a constant for all entries, and at that point VIEC will be less efficient than MBIE at solving this task. I imagine that figuring out a near optimal demand matrix may be almost as hard as solving the MDP, in which case VIEC will at best be tied with MBIE for SixArms. The reason for bringing up this example is to highlight that whether reward traps are good or bad is completely domain dependent. In some MDPs you want to be attracted to reward because MBIE will learn faster, and in other MDPs the rewards can be traps which will slow MBIE down. Avoiding reward traps is only consistently a bad thing if your goal is pure exploration. But if the goal was pure exploration, then R-MAX/MBIE are the wrong algorithms to compare to, as their goal is not pure exploration. Like I mentioned in the review, perhaps the exploration phase of E^3 would be a better comparison. But an even easier baseline to compare to is R-MAX/MBIE without giving them the reward signal i.e. letting the real reward always be 0. This is a very simple thing to do if you want pure exploration, and this completely avoids reward traps. My point is that either you should mention that reward traps being good or bad are problem dependent if you want to compare RL performance, or if you want to compare pure exploration then you should simply give reward of 0 to R-MAX/MBIE which basically means reward traps are no longer an issue - this should be a simple change as R-MAX/MBIE with a reward of 0 will still suffer from distance traps in the tower MDPs and I believe all your analyses still hold. The main issue for me is still that there are very large theoretical and practical challenges to this approach. The main practical challenge is the computation cost. You mention an approximate algorithm that works well in practice, and so I hope you include that in the final copy, as that would help introduce a method that could potentially be extended to work in practice. The main theoretical challenges of learning a dynamics model simultaneously and where to get a demand matrix in the first place are difficult and should be for future work. That said, the potential of the demand matrix is enough for me to increase my score.

Reviewer 2



Although the estimate of the computational order is limited to some simple problems, the authors successfully demonstrate when the existing approaches such as epsilon-greedy, R-MAX and interval estimation based exploration strategies result in poor performance compared with the proposed algorithm. The study is quite thought provoking and I believe it would stimulate the future research on this area. ------------------------------------------------------------------------- After reading the other reviewers' comment, and the concerns how to determine the demand matrix is fair. SoI will lower my score from 9 to 8.

Reviewer 3



Originality: it is difficult for me to assess the level of novelty of the ideas proposed in the work proposed in this paper. It seems to me that addressing the exploration problem using an additional planning strategy is close to what has been done in the Bayesian RL literature, in particular, Bayesian approaches doing planning in a Belief-augmented MDP. Quality: it is difficult for me to assess the quality of the proposed approach (at least for its exploration performance). I could not see any empirical evaluation, nor clear theoretical statements (section 4. provides some more formal analyses, but without any claims (I mean Lemma, or Theorem) coming with proofs. Clarity: despite the paper lists contributions at the beginning, I personally found the structure of the paper a bit complicated to follow. In particular, I found Section 4. strangely written: it proposes some form evaluations, but without structuring it in a sequence of Lemma and proofs. Sometimes, the paper develops some intuitions starting from imaginary MDPs, for instance "For example, in an MDP with two states {s1, s2} [...] , a demand D[s1, a] > 1 can never be satisfied and will lead to an infinite exploration cost." These types of arguments could be formalised into lemmas or propositions. Significance: the paper may be of interest for researchers interesting in sample efficiency of RL algorithm. However, the paper should emphasize more (and, perhaps, more quantitatively, including empirical evidence) how the proposed algorithm positions itself of the proposed approach regarding other algorithms. *** Post feedback *** I have updated the overall score according to the feedback provided by the authors.

[Author Response · NeurIPS 2019]

We thank all reviewers for providing insightful comments and helpful suggestions. We have revised some parts of our
paper accordingly to improve the clarity. The following are our responses to some specific topics.

**Regarding adding a PAC-style analysis for VIEC using learned models (Reviewer 1).** Including such an analysis
is indeed good for the sake of completeness, but we feel that it also misses the main points of this paper: (1) an
exploration strategy with PAC guarantees (R-MAX, MBIE, etc.) can still be far from optimal in terms of exploration
cost; (2) explicit planning has remarkable potentials in improving it. Such potentials can only be shown by comparing
the exploration costs of the best exploration scheme (the optimal one for the true MDP) and the ones actually taken by
the existing algorithms. PAC-style analysis stresses the worst cases, and thus cannot be used to show such potentials.
This is a little like guaranteeing that one can run 100m in at most 1 minute, but it does not say how (or even whether)
one can achieve 9.6s (which is the main concern of this paper).

It is also very difficult (if not impossible) to use a PAC-style analysis to expose the weaknesses of algorithms such as
distance / reward traps, because such an analysis usually occurs at an abstract level and does not show exactly when and
how wrong decisions are made during exploration.

Therefore, while the PAC-style analysis is an interesting suggestion and can surely bring some more insight, we do not
consider it essential for the purpose of this paper.

**Regarding the reward traps, exploration vs exploitation, and the Six Arms example (Reviewer 1).** Although
algorithms like R-MAX and MBIE are conceptually designed for trading-off exploration and exploitation, in reality
they achieve it by doing "pure" exploration first and then converging to some policy at certain stage. As Reviewer 1
mentioned in the Six Arms example, MBIE first explores and tries out different arms, then stops exploration and starts
exploiting the best arm after it has sufficient evidence for confirming the optimality of that arm with a high confidence.
This is a clear example of exploration-then-exploitation behaviour with exactly one phase change in the process.

As we have discussed in Section 3.1, the minimum sample size for confirming the optimality of a policy / action is given
by the Hoeffding's or Chernoff's inequalities, which implicitly result in a fixed demand matrix for a given MDP. Given
the same demand matrix, the optimal exploration scheme never chooses actions more than necessary (by definition),
while MBIE can be distracted by immediate rewards and eventually choose suboptimal actions more often than needed
(which we call reward traps). Therefore, no matter how the domain is designed, MBIE always needs more steps to
fulfil a demand matrix (i.e. finish its exploration phase) than the optimal exploration scheme (unless the demand matrix
happens to match the actual behaviour of MBIE exactly, in which case the costs are the same).

Even in the case where the total reward is of primary concern (and thus "balancing" exploration and exploitation is
important), being lured to the reward traps only prolongs the exploration phase of MBIE, which leads to strictly less
total reward in the long run. Therefore, what we called reward traps are still traps in such cases.

We understand that this part can be somewhat tricky (especially to readers accustomed to traditional concepts of
exploration). More discussion is added to the final version to help clarify our ideas.

**Regarding the computational cost (Reviewer 1).** We agree that the computational cost for VIEC is too high to be
of practical use. However, an efficient approximation is practical, since a large portion of the demand space is not
crucial in deciding the (near-)optimal scheme. We have a simple implementation of the Monte-Carlo approximation
that can achieve O(nmd) in tower MDPs like the optimal scheme (albeit with a worse constant factor) which computes
much faster than VIEC. This, of course, is still not as fast as heuristic based algorithms and could benefit from further
improvement. This will be our future work.

**Regarding the lack of numerical experiments (Reviewer 2 & 3).** We agree that including a section of empirical
results might make the paper more complete. In fact, it is actually more difficult to visualise the exploration behaviours
through empirical results than through the analysis done in Section 4. If we only provide the exploration cost curves, it
will not provide more useful information than the theoretical results shown in Table 1 of Section 4. We consider that
providing more details about how existing algorithms behave during exploration and why some decisions are bad is
more helpful for readers to understand the weaknesses of those algorithms as well as the potential of explicit planning,
hence the focus on the theoretical analysis in this paper.

**Regarding the relationship with Bayesian RL (Reviewer 3).** The resemblance between the two is a little superficial.
Although both do planning in augmented MDPs, the objectives of planning, the augmented state spaces, and the
definitions of optimality are all different. In the context of this paper, Bayesian RL is essentially yet another family of
heuristic-based methods. It chooses actions that are (approximately) Bayesian optimal, which are not guaranteed to be
optimal with respect to exploration costs. In fact, since Bayesian methods take immediate rewards into consideration,
they are prone to reward traps just like the IE family elaborated in Section 4.5. The ones that use discounts in planning
are also prone to distance traps.

[Meta-Review · NeurIPS 2019]

All the reviewers and I liked the paper. Congratulations!